

# Long-term continuously monocropped peanut significantly changed the abundance and composition of soil bacterial communities

Mingna Chen[1,2], Hu Liu[1], Shanlin Yu[2], Mian Wang[2], Lijuan Pan[2], Na Chen[2], Tong Wang[2], Xiaoyuan Chi[2] and Binghai Du[1]

[1] College of Life Sciences, Shandong Key Laboratory of Agricultural Microbiology, National Engineering Laboratory for Efficient Utilization of Soil and Fertilizer Resources, Shandong Agricultural University, Taian, China

[2] Shandong Peanut Research Institute, Qingdao, China

## ABSTRACT

Soil sickness is the progressive loss of soil quality due to continuous monocropping. The bacterial populations are critical to sustaining agroecosystems, but their responses to long-term peanut monocropping have not been determined. In this study, based on a previously constructed gradient of continuous monocropped plots, we tracked the detailed feedback responses of soil bacteria to short- and long-term continuous monocropping of four different peanut varieties using high-throughput sequencing techniques. The analyses showed that soil samples from 1- and 2-year monocropped plots were grouped into one class, and samples from the 11- and 12-year plots were grouped into another. Long-term consecutive monocropping could lead to a general loss in bacterial diversity and remarkable changes in bacterial abundance and composition. At the genera level, the dominant genus *Bacillus* changed in average abundance from 1.49% in short-term monocropping libraries to 2.96% in the long-term libraries. The dominant species *Bacillus aryabhattai* and *Bacillus funiculus* and the relatively abundant species *Bacillus luciferensis* and *Bacillus decolorationis* all showed increased abundance with long-term monocropping. Additionally, several other taxa at the genus and species level also presented increased abundance with long-term peanut monocropping; however, several taxa showed decreased abundance. Comparing analyses of predicted bacterial community functions showed significant changes at different KEGG pathway levels with long-term peanut monocropping. Combined with our previous study, this study indicated that bacterial communities were obviously influenced by the monocropping period, but less influenced by peanut variety and growth stage. Some bacterial taxa with increased abundance have functions of promoting plant growth or degrading potential soil allelochemicals, and should be closely related with soil remediation and may have potential application to relieve peanut soil sickness. A decrease in diversity and abundance of bacterial communities, especially beneficial communities, and simplification of bacterial community function with long-term peanut monocropping could be the main cause of peanut soil sickness.

Corresponding authors
Xiaoyuan Chi, chi000@126.com
Binghai Du, dubhai@126.com

## INTRODUCTION

Soil sickness is the progressive loss of soil quality due to continuous monocropping and results in the reduction of crop yield and quality, as well as a prevalence of soil-borne diseases (*Huang et al., 2013*; *Van der Putten et al., 2013*). It is a major problem in agriculture ecosystems all over the world, and has been reported for many types of crops, including food (e.g., rice, wheat, corn, soybean, and peanut), economic (e.g., sugarcane and tobacco), vegetable (e.g., cucumber and eggplant), and medicinal crops (e.g., *Rehmannia*, ginseng, and *Angelica*) (*Liu et al., 2012*; *Gentry, Ruffo & Below, 2013*; *Huang et al., 2013*; *Wu et al., 2015*). Monocropping is considered unsustainable in agricultural systems; however, modern agricultural practices are often characterized by monocropping (*Cook, 2006*).

Based on previous reports, there are four main factors contributing to soil sickness: disorder in physicochemical soil properties, production, and accumulation of autotoxins, imbalance of soil microbial communities and change in soil enzyme activity (*Huang et al., 2013*; *Zhou et al., 2018*). Soil microorganisms that are critical to many soil biological, chemical, and physical processes such as soil structure formation, mineral nutrition cycling, organic matter turnover and toxin accumulation or removal, are considered to be key drivers of terrestrial ecosystems (*Bever, Platt & Morton, 2012*; *Blagodatskaya & Kuzyakov, 2013*). Alteration of soil microbial communities can change the function performed by the communities and then feedback on plant growth and health (*Bever, Platt & Morton, 2012*; *Zhou, Liu & Wu, 2017*). In addition, the soil microbial community can serve as a sensitive bioindicator of soil health due to its quick response to environmental changes and close relationship with soil conditions and land management (*Sharma et al., 2010*). Consequently, understanding how soil microbial communities are affected by continuous monocropping is necessary to provide insights into soil sickness.

Bacterial populations are most abundant and diverse in soil, and the interactions between soil, bacteria, and plants in root-related environments play key roles in soil fertility, sustainability, and plant quality (*Chaparro et al., 2012*). Many bacterial taxa have been identified as biocontrol agents against soil-borne pathogens and play key roles in promoting plant growth, and some soil bacteria also have been reported as plant pathogens (*Compant, Clément & Sessitsch, 2010*; *Santoyo, Orozco-Mosqueda & Govindappa, 2012*; *Buttimer et al., 2017*). Increasing evidence indicates that the soil bacterial communities can be shaped by plants through secretion of root exudates (*Doornbos, Van Loon & Bakker, 2012*). Modifications in soil microbe populations induced by peanut (*Arachis hypogaea* L.) root exudates, rather than direct allelopathy, could contribute to peanut soil sickness (*Li et al., 2014a*). Additionally, recent studies suggested that accumulations of microbial pathogens at the expense of plant-beneficial microorganisms in the soil are likely explanations for yield declines as a consequence of consecutive monocropping (*Chen et al., 2012*; *Li et al., 2014b*; *Xiong et al., 2015*). Therefore, clarifying changes in soil bacterial community properties in continuous monocropping systems should be helpful for developing practices to relieve soil sickness in agricultural production.

Peanut, an important oil and economic crop worldwide, is very adaptable to climatic conditions and grows in tropical, subtropical, and warm temperate climate regions across

the world. Due to limitations of arable land and requirements for developing regional agro-industrialization, large-scale monocropping of peanut is common in China (*Chen et al., 2016*). Research indicates that consecutive peanut monocropping has caused a decline in yield and quality and increases in disease pressures (*Wang & Chen, 2005*). Early studies showed that soil diversity and abundance of bacterial communities changed with continuous peanut monocropping according to phospholipid fatty acid (PLFA), denaturing gradient gel electrophoresis and library analyses (*Li et al., 2012*; *Chen et al., 2014*; *Liu et al., 2015*). Our earlier study also indicated that the balance of soil bacterial communities was disturbed during three years of continuous monocropping (*Chen et al., 2014*). However, the specific characteristics of the soil bacterial community and the changes of soil bacterial structure and composition in response to long-term peanut monocropping are unclear.

In this study, based on a gradient that we previously constructed of continuously monocropping in a peanut field, we analyzed and compared responses of root soil bacterial communities of four peanut varieties to monocropping for 1, 2, 11, and 12 years, using high-throughput sequencing techniques. This study aims to investigate bacterial community dynamics succession under long-term peanut monocropping based on monocropping gradient experiment plots with a consistent background. The aims of this study were to (i) determine the change characteristics of the soil bacterial community and the influences of peanut varieties on the dynamics of the bacterial community, under long-term continuous monocropping of peanut, and (ii) identify the key bacteria taxa related to peanut soil sickness.

# MATERIALS AND METHODS

## Field experiment and soil sampling

The field experimental site was set up in Laixi experimental farm at the Shandong Peanut Research Institute, Qingdao, China (36°50′N, 120°31′E). A gradient of consecutive monocropped peanut experiment plots that we previously developed was used. Independent pools were applied, with the size of each pool being 4 m long, 1.5 m wide, and 1 m high. We collected soil from the plow layer of the cultivated land that had previously been planted with a wheat–maize rotation, thoroughly mixed, and then added to the pools. In order to construct the gradient of continuous monocropping plots, sweet potatoes were rotated with peanut in some plots during the experiment. Peanuts were planted in May each year and harvested in October. All field management, including the planting pattern and the use of water and fertilizer were consistent among different pools. After harvest, the plots lay fallow until the next planting season. The distance between rows was 16.5 cm and the ridge width was 70 cm, with two peanut seeds per hole. Before planting, about 300 kg ha$^{-1}$ of urea, 750 kg ha$^{-1}$ of calcium magnesium phosphate, and 225 kg ha$^{-1}$ of potassium chlorate were applied as fertilizer. The weeds were controlled using glyphosate (41% active ingredient in 3 L ha$^{-1}$) before planting. Plastic film mulching was applied during cultivation. In addition, to minimize the influence of other factors including variety on the measured indexes, the planting position of the same peanut variety was fixed during the continuous cropping period. The longest-running pools have now been continuously monocropped with peanuts for 16 years.

According to our previous study (*Jiao et al., 2015*), we selected four peanut varieties with distinct responses to monocropping to analyze their soil bacterial community structure in plots monocropped for 1, 2, 11, and 12 years. The four varieties included two large fruit varieties, Huayu 917 and Huayu 50, and two small fruit varieties, Huayu 26 and Huayu 20. Based on previous testing of their yield-related indexes, Huayu 20 was the most sensitive to long-term monocropping, Huayu 917 was intermediate, and Huayu 26 and Huayu 50 were tolerant. There were three replicates for each treatment. The bulk soils were collected at the full-bloom stage and five randomly selected replicate test plants were used for each sample. The soils around individual plants were sampled using a soil probe (1.5 cm diameter) at 5–10 cm soil depth, and at distances of 3–5, 8–10, and 13–15 cm from the main root. The root zone soils were then mixed together. These distances were chosen because the roots at this depth and these distances were relatively abundant and the soil microbial community around the peanut root system could be well characterized (*Chen et al., 2012*). Physico-chemical properties of pH, organic matter, and available nitrogen (N), phosphorus (P), and potassium (K) in the soil samples were determined using routine methods (*Lu, 1999*). Samples Y1, Y2, Y11, and Y12 were for monocropping of 1, 2, 11, and 12 years, respectively; H20, H26, H50, and H917 were samples for four peanut varieties with distinct responses to monocropping.

## DNA extraction, PCR, and high-throughput sequencing

For each soil sample, total DNA was extracted with the Power Soil DNA Isolation Kit (MoBio Laboratories Inc., Carlsbad, CA, USA) following the manufacturer's instructions. The concentration and purity of the DNA was checked by electrophoresis on 1.0% (w/v) agarose gels. Soil bacterial communities were analyzed with amplicon sequencing on an Illumina HiSeq platform. Two primers, 515F (5′-GTGCCAGCMGCCGCGGTAA-3′) and 907R (5′-CCGTCAATTCCTTTGAGTTT-3′), for the V4–V5 region of the 16S rRNA gene were applied. Both the forward and reverse primers had a 6-bp barcode unique to each sample. Each soil sample was independently amplified in triplicate, and the triplicate PCR reaction products for each sample were pooled and then all products were purified using a GeneJET$^{TM}$ gel extraction kit (Thermo Scientific, Waltham, USA). The purified amplicons from each sample were pooled in equimolar concentrations and the mixture was then sequenced on an Illumina HiSeq platform at Novogene Co. Ltd, Beijing, China.

## Sequencing data processing and statistical analysis

The paired-end reads from the original DNA fragments were merged with FLASH software (*Edgar, 2013*). The QIIME 1.7.0 software package was used to quality-filter and process the raw sequencing reads (*Caporaso et al., 2010*). The UCLUST method was used to delineate the operational taxonomic units (OTUs) at a threshold of 97% identity (*Caporaso et al., 2010*). A representative sequence for each OTU was taxonomically classified using the Silva Database (https://www.arb-silva.de/) based on Mothur algorithm (*Quast et al., 2013*). The OTU abundance data were normalized using a standard sequence number corresponding to the sample with the least sequences. The output normalized data were used for the subsequent analyses.

To assess the bacterial alpha-diversity of each sample, analyses including rarefaction curves, Shannon index, and Chao1 index were performed using QIIME (Version 1.7.0) and displayed with R software (Version 2.15.3). Beta diversity analyses were used to estimate sample differences in bacterial community compositions. In order to analyze the influence of peanut monocropping on soil bacterial community composition, heatmap analysis was performed with R software. Cluster analysis was preceded by principal component analysis (PCA), which was applied to reduce the dimensions of the original variables using the FactoMineR package and ggplot2 package in R software (Version 2.15.3). Principal coordinate analysis (PCoA) was performed to obtain principal coordinates and visualize the complex multidimensional data. The PCoA analysis was performed using the WGCNA, stat, and ggplot2 packages in R software (Version 2.15.3). Unweighted pair-group method with arithmetic means (UPGMA) clustering was performed as a type of hierarchical clustering method to interpret the distance matrix using average linkage and was conducted using QIIME software (Version 1.7.0). Taxa with significantly diverse abundance with long-term monocropping at different levels were further investigated using the Metastats method and $t$-test using R software (Version 2.15.3). Tax4Fun functional prediction was achieved by the nearest neighbor method based on the minimum 16S rRNA sequence similarity. The data set was deposited in the NCBI-Sequence Read Archive with the submission accession number PRJNA559575.

## RESULTS

### Physico-chemical properties of soil

Physico-chemical properties of pH, organic matter, and available N, P, and K in the peanut monocropping soil were tested and analyzed (Table S1). The soil pHs in samples from 2-year monocropped plots (8.30–8.59) were slightly higher than those from 1-year monocropped plots (7.20–8.05), but were obviously lower in samples from 11- and 12-year monocropped plots (6.46–6.92). The contents of available P were lower in long-term (15.12–21.12 mg kg$^{-1}$) than in short-term monocropping samples (39.37–48.58 mg kg$^{-1}$). The contents of available K were also lower in long-term (59.98–86.13 mg kg$^{-1}$) than in short-term monocropping samples (99.42–120.67 mg kg$^{-1}$). However, contents of organic matter and available N showed no obvious changes with long-term peanut monocropping.

### Characteristics of sequencing data

In order to compare the soil bacterial community structure and composition of the four peanut varieties to short- and long-term monocropping periods, a total of 48 16S rRNA gene libraries were analyzed using high-throughput sequencing. The sampling variables, including four peanut varieties and four monocropping periods, are shown in Table 1. A total of 2,541,421 quality-filtered sequences obtained from the 48 samples, ranging within 37,605–68,117 with an average length of 373 bp, resulted in identification of a total of 9028 OTUs applying a 3% sequence dissimilarity cutoff. After data normalization, 1,805,040 quality-filtered sequences affiliated with 8856 OTUs were obtained. The bacterial complexity of the 48 samples was estimated on the basis of alpha-diversity (OTU number, Chao1 index, and Shannon index) and showed relatively higher bacterial diversity in

**Table 1  Overview of soil samples obtained from plots that were monocropped for different periods.** The table shows the number of quality sequences and the indexes for α-diversity. Operational taxonomic units (OTUs) are defined at 97% sequence similarity.

| Sample name | Monocropping years | peanut varieties | Sequenced library No. | Total filtered quality sequences | Richness estimates | | Diversity estimates |
|---|---|---|---|---|---|---|---|
| | | | | | OTUs | Chao1 | Shannon |
| Y1.H20 | 1 | Huayu 20 | 3 | 211427 | 5132 ± 327 | 4296 ± 727 | 9.92 ± 0.22 |
| Y1.H26 | 1 | Huayu 26 | 3 | 201027 | 5262 ± 105 | 4287 ± 118 | 10.10 ± 0.06 |
| Y1.H50 | 1 | Huayu 50 | 3 | 217300 | 5187 ± 66 | 4189 ± 86 | 10.02 ± 0.04 |
| Y1.H917 | 1 | Huayu 917 | 3 | 209336 | 5212 ± 33 | 4439 ± 432 | 10.01 ± 0.05 |
| Y2.H20 | 2 | Huayu 20 | 3 | 207920 | 5074 ± 69 | 4303 ± 444 | 9.93 ± 0.04 |
| Y2.H26 | 2 | Huayu 26 | 3 | 197796 | 5083 ± 139 | 4075 ± 166 | 9.93 ± 0.11 |
| Y2.H50 | 2 | Huayu 50 | 3 | 220390 | 5118 ± 23 | 4445 ± 274 | 9.99 ± 0.03 |
| Y2.H917 | 2 | Huayu 917 | 3 | 227617 | 5462 ± 176 | 4839 ± 517 | 10.14 ± 0.08 |
| Y11.H20 | 11 | Huayu 20 | 3 | 219846 | 4619 ± 181 | 3756 ± 212 | 9.70 ± 0.15 |
| Y11.H26 | 11 | Huayu 26 | 3 | 196398 | 4778 ± 71 | 3852 ± 115 | 9.83 ± 0.06 |
| Y11.H50 | 11 | Huayu 50 | 3 | 196570 | 4693 ± 39 | 3742 ± 51 | 9.83 ± 0.00 |
| Y11.H917 | 11 | Huayu 917 | 3 | 229162 | 5064 ± 67 | 4428 ± 369 | 9.90 ± 0.05 |
| Y12.H20 | 12 | Huayu 20 | 3 | 210578 | 5036 ± 62 | 4056 ± 68 | 9.93 ± 0.05 |
| Y12.H26 | 12 | Huayu 26 | 3 | 212672 | 5120 ± 56 | 4294 ± 363 | 9.96 ± 0.02 |
| Y12.H50 | 12 | Huayu 50 | 3 | 199615 | 4971 ± 63 | 4209 ± 321 | 9.91 ± 0.07 |
| Y12.H917 | 12 | Huayu 917 | 3 | 197203 | 5049 ± 233 | 4017 ± 328 | 9.97 ± 0.07 |

samples from the 1- and 2-year compared to the 11- and 12-year monocropped plots (Table 1). Rarefaction curves (Fig. S1) and indices of richness and diversity including all samples' Chao1 and Shannon indexes (Table 1) tended to approach a saturation plateau, indicating that the majority of bacterial diversity was recovered by the surveying effort.

## Bacterial community structure and composition

In total, 99.73% of the identified 8856 OTUs were related to bacteria, and the other 24 OTUs accounting for 0.15% of the total quality-filtered sequences were affiliated with Archaea. There were 41 bacteria phyla detected across all samples and 0.73% of bacterial sequences were unclassified at the phylum level. The top 10 phyla accounted for 95.49% of the total sequences and 79.67% of the total OTUs. The dominant phyla across all samples were Proteobacteria, Actinobacteria, Acidobacteria, and Chloroflexi, accounting for 30.71, 21.59, 18.04, and 7.69% of the total sequences, respectively, and 26.61, 7.43, 6.49, and 10.77% of the total 8856 OTUs, respectively. The phyla Gemmatimonadetes, Firmicutes, Planctomycetes, Nitrospirae, Bacteroidetes, and Thermomicrobia were relatively abundant and diverse, accounting for 4.66, 3.75, 3.13, 2.96, 1.62, and 1.36% of the total sequences, respectively, and 3.42, 4.41, 14.76, 0.61, 3.97, and 1.19% of the total OTUs, respectively (Fig. S2).

At the class level, 92 bacteria taxa were identified. The dominant 19 taxa (relative abundance > 1%) accounted for > 88.43% of the sequences in each sample. The most abundant taxa (relative abundance > 5%) were Acidobacteria (16.19%), Alphaproteobacteria (10.11%), Betaproteobacteria (9.48%), Thermoleophilia (9.22%),

Actinobacteria (7.31%), Deltaproteobacteria (5.77%), and Gammaproteobacteria (5.29%). The other relatively abundant taxa included Gemmatimonadetes, Bacilli, Nitrospira, MB-A2-108, Planctomycetacia, Acidimicrobiia, Anaerolineae, KD4-96, Holophagae, Thermomicrobia, Sphingobacteriia, and TK10, each accounting for 1.05–4.66% of the total sequences (Fig. 1A).

At the genus level, 494 taxa were detected, accounting for 31.41% of the total sequences; and the top 50 taxa accounted for 75.04% of the total identified sequences at the genus level. The dominant genera (relative abundance > 1%) were *Bacillus*, *Gaiella*, *Mizugakiibacter*, and *Streptomyces*, accounting for 2.22, 1.79, 1.35, and 1.01% of the total sequences, respectively. Relatively abundant genera were *Sphingomonas* (0.90%), *Haliangium* (0.89%), *Gemmatimonas* (0.80%), *Bryobacter* (0.76%), *Nocardioides* (0.74%), *Arthrobacter* (0.71%), *Steroidobacter* (0.68%), *Solirubrobacter* (0.66%), and *Microvirga* (0.57%) (Fig. 1B).

## Bacterial community variation at different levels with monocropping time

In order to illuminate the bacterial community variation across long-term monocropping time of peanut, UPGMA, PCA, and PCoA were used to cluster the bacterial communities within soil samples. Overall, the UPGMA tree showed that replicate samples were grouped together, and soil samples with similar monocropping time but different peanut varieties were clustered into a group (Fig. 2). Both PCA and PCoA also showed obvious clustering of bacterial communities based on monocropping time (Fig. 3, Fig. S3). Samples from 1- and 2-year monocropping plots were grouped into one class, and samples from 11- and 12-year plots were grouped into another. However, the cluster distance was shorter among samples from 1- and 2-year than among samples from 11- and 12-year monocropping plots. These analyses indicated that bacterial communities were significantly influenced by the monocropping period of peanut, and showed more obvious changes under long-term monocropping.

The heatmap analyses showed that soil bacterial community composition was significantly diverse at different levels between short- and long-term monocropping. At the phylum level, among the 41 bacteria taxa identified, nine taxa showed an obvious decrease in abundance with monocropping time and five taxa showed an obvious increase in abundance (Fig. S4, Fig. 4). The average abundance of Acidobacteria accounted for 20.30 and 19.83% in the 1- and 2-year libraries, respectively, and decreased to 15.74 and 16.28% in the 11- and 12-year libraries. The phyla Planctomycetes, Nitrospirae, and Bacteroidetes were relatively abundant in the 1- and 2-year libraries, accounting for 3.12, 2.96, and 1.62% of the total sequences, respectively, and correspondingly decreased in the 11- and 12-year libraries by 20.23, 17.47, and 27.81%. Additionally, the taxa Armatimonadetes, Latescibacteria, JL-ETNP-Z39, Thermotogae, and Caldiserica, for which relative abundances were <1%, also had clear decreasing tendencies. In general, decreases in their abundance were accompanied by decreases in their diversity (Table S2). In contrast, the abundance of some taxa increased with monocropping time. The average abundance of Gemmatimonadetes, Firmicutes, and Elusimicrobia accounted for 3.91, 2.65, and 0.16% in the short-term monocropping libraries, respectively, and correspondingly increased to 5.40, 4.84, and

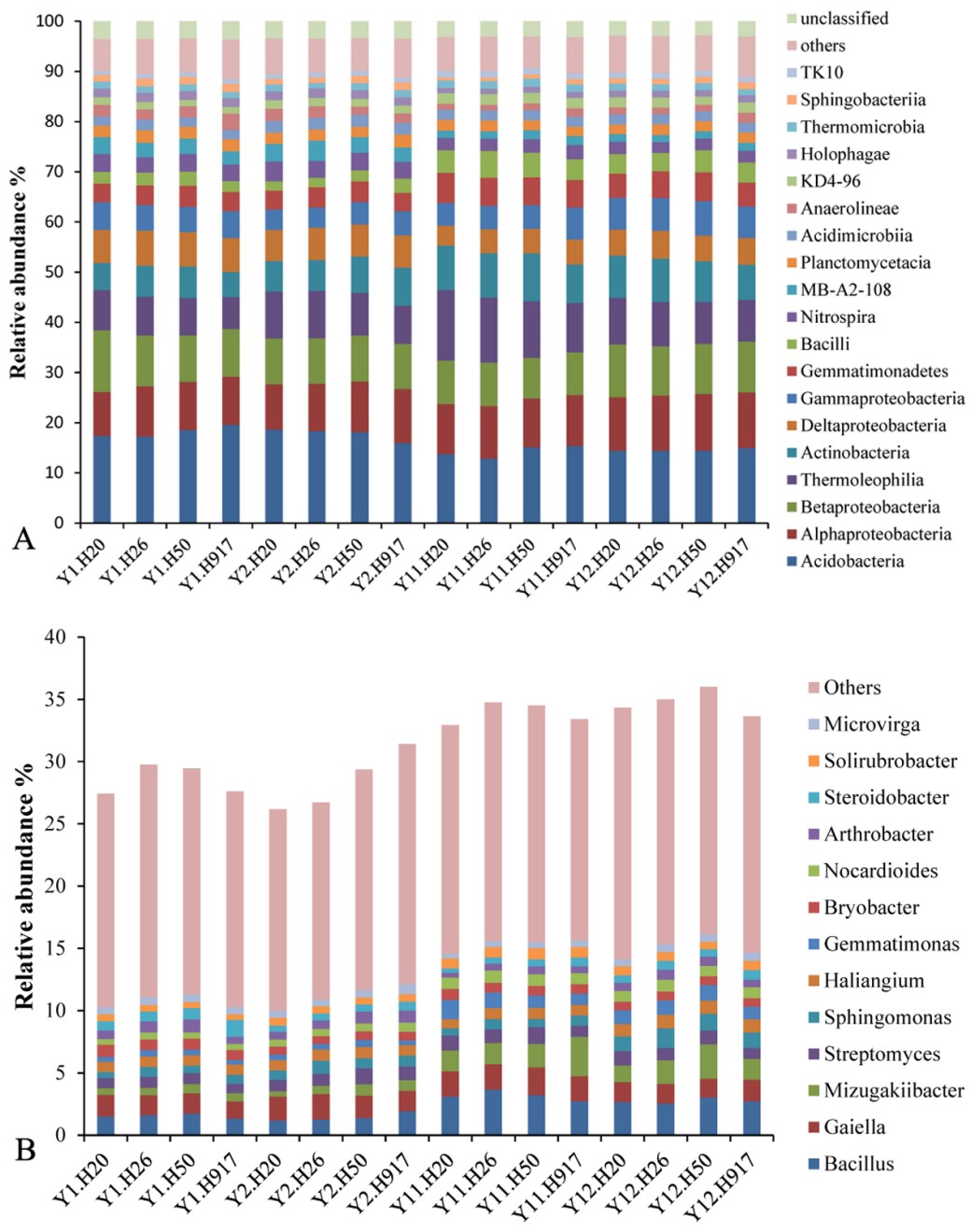

**Figure 1** Overall abundance distribution of bacteria at the (A) class and (B) genus levels from soil that was monocropped with peanut.

0.21% for the long-term. The other taxa with increasing abundance all had relatively low abundance, accounting for 0.01–0.04% of the total sequences, and included Parcubacteria and Chlamydiae. The diversity of the three relatively abundant taxa (Gemmatimonadetes, Firmicutes, and Elusimicrobia) did not show obvious changes with monocropping time,

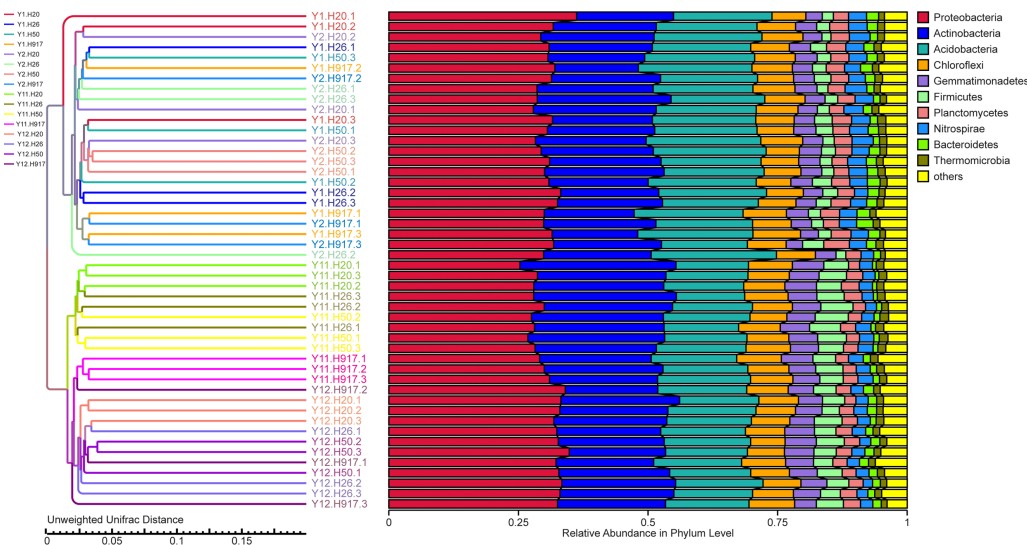

**Figure 2** The UPGMA tree showing the clusters for bacterial communities in short- and long-term monocropped soils based on the unweighted UniFrac value.

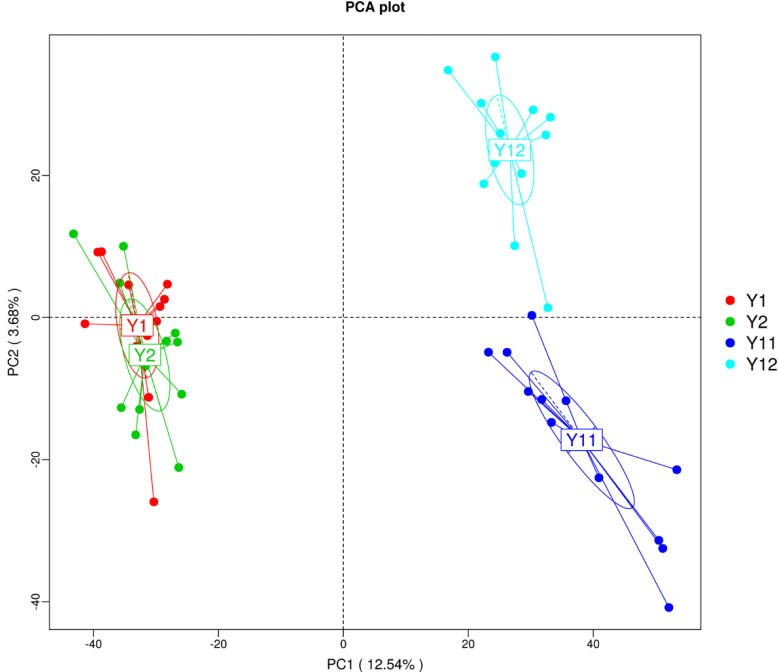

**Figure 3** Principal component analysis (PCA) of bacterial distributions in short- and long-term monocropped peanut soils for different peanut varieties.

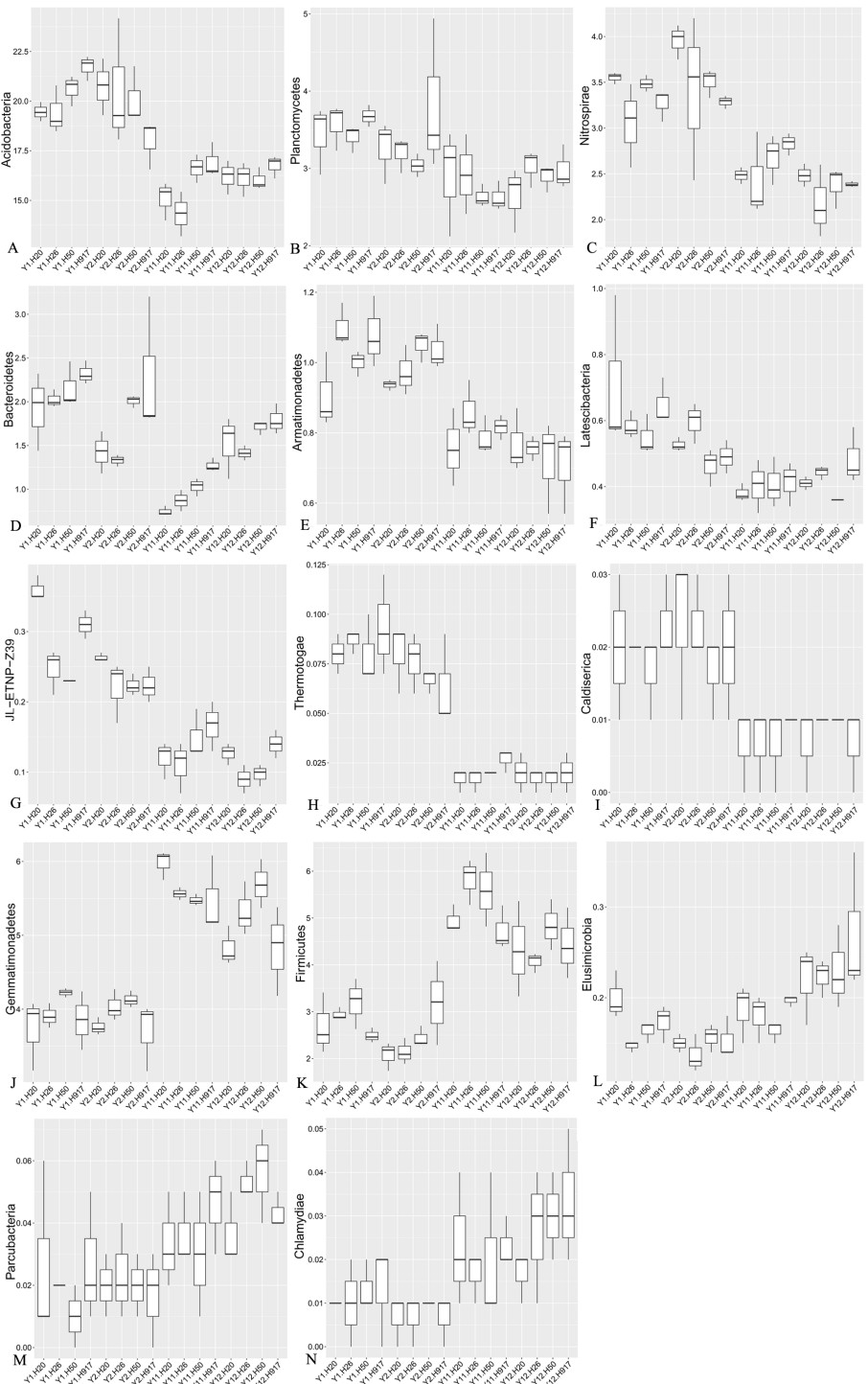

**Figure 4 Relative abundance of significantly changed taxa at phylum level in plots that were monocropped with peanuts for different periods (A–N).**

but that of the other low-abundance taxa tended to increase with monocropping time (Table S2).

Analysis of the microbial community at lower taxonomic levels (genus or species) can provide better phylogenetic resolution than at higher taxonomic levels (*Ho et al., 2016*). At the genera level, the top 50 most-abundant taxa accounted for 75.04% of the total identified sequences that were affiliated with 494 taxa. The abundance distributions of the top 50 genera in each sample were presented in a heatmap (Fig. 5). Some taxa showed obvious changes with monocropping time (Figs. S5 and S6). The dominant genera *Bacillus* (2.22%) and *Mizugakiibacter* (1.35%) increased their average abundance from 1.49 and 0.67%, respectively, in the 1- and 2-year libraries to 2.96 and 2.02% in the 11- and 12-year libraries. However, the number of OTUs affiliated with the genera *Bacillus* (13–14 OTUs) and *Mizugakiibacter* (four OTUs) in each sample was relatively low and changed little. Seven genera (*Gemmatimonas*, *Marmoricola*, *Candidatus* Solibacter, *Tumebacillus*, *Jatrophihabitans*, *Sporosarcina*, and *Pseudolabrys*) accounted for 0.17–0.80% of the total sequences, and increased their average abundance by >90%. The genera *Nocardioides*, *Solirubrobacter*, and *Reyranella* accounting for 0.21–0.74% of the total sequences also showed increased abundance with monocropping time, with increase rates of 26.44–57.20%. Some taxa showed an opposite trend, with abundance decreasing with monocropping time. They were *Steroidobacter*, *Candidatus Entotheonella*, *Nitrosospira*, *Pirellula*, *Piscinibacter*, *Ramlibacter*, *Lysobacter*, *Skermanella*, and *Planctomyces*, which accounted for 0.20–0.68% of the total sequences, and the abundance decrease rate ranged within 12.86–51.13%. Overall, abundance of these genera began to decline in the 2-year monocropping libraries, and decreased more for the long-term libraries.

Most sequences, accounting for 91.54% of the total sequences, were not classified at the species level, but some taxa among the identified species also obviously changed with monocropping time (Fig. S7, Fig. 6). Among the top 10 most-abundant species, *Bacillus aryabhattai* and *Bacillus funiculus* accounting for 0.58 and 0.42% of the total sequences, respectively, showed increased abundance in long- compared to short-term monocropping libraries. The increase rate of these two taxa reached 116.23 and 177.70%, respectively. The relatively abundant taxa of *Bacillus luciferensis*, *Tumebacillus ginsengisoli*, *Bacillus decolorationis*, and *Streptacidiphilus luteoalbus*, accounting for 0.01–0.10% of the total sequences, increased their relative abundances by > 118%. The taxa *Paenibacillus alginolyticus* and *Streptosporangium violaceochromogenes* also presented increased abundance with increase rates ranging within 52.07–84.49%, but the change trends differed among peanut varieties. The species *Paenibacillus alginolyticus* and *Streptosporangium violaceochromogenes* showed no obvious changes in abundance in Huayu 917 and Huayu 50 libraries, respectively. In contrast to the above taxa, *Chitinophaga ginsengihumi*, *Lysobacter yangpyeongensis*, and *Phyllobacterium myrsinacearum*, accounting for 0.03–0.05% of the total sequences, showed obvious decreases in abundance and the abundance decrease rate ranged within 32.57–61.59%.
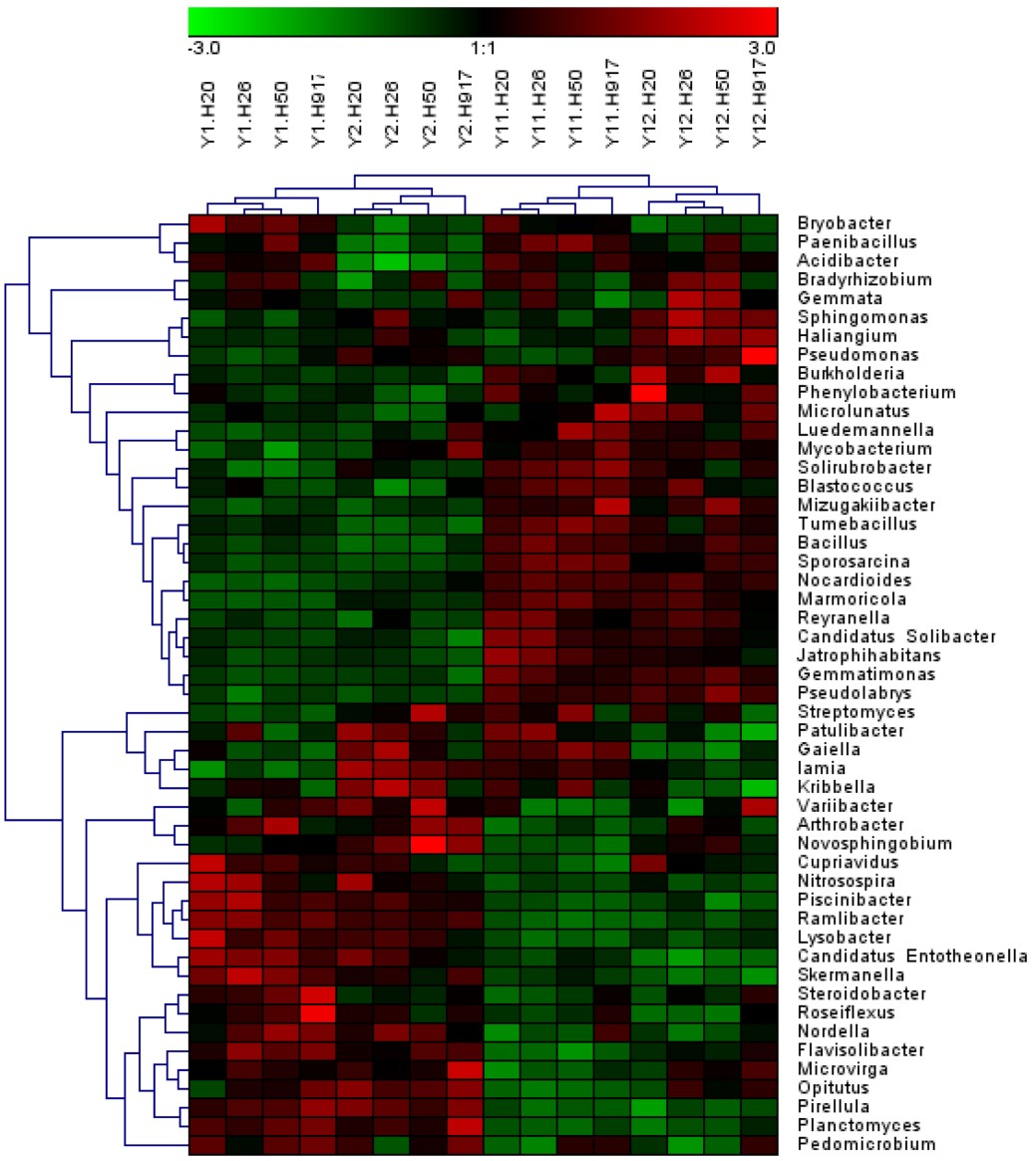

**Figure 5** Heatmap presenting the distribution of the top 50 taxa at genus level in 16 monocropped peanut plots.

## Prediction of bacteria community functions

The predominant category of the predicted functional genes was affiliated with metabolism (48.71%), followed by genetic information (21.03%), environmental information processing (12.89%), and cellular processes (7.62%) (Fig. S8). The PCA was used to cluster the predicted functional pathway within soil samples at three KEGG levels. The analyses at KEGG levels 2 and 3 and KEGG ortholog (KO) level all showed similar clusters to that of bacterial community structure (Fig. 7, Fig. S9). Samples from 1- and 2-year

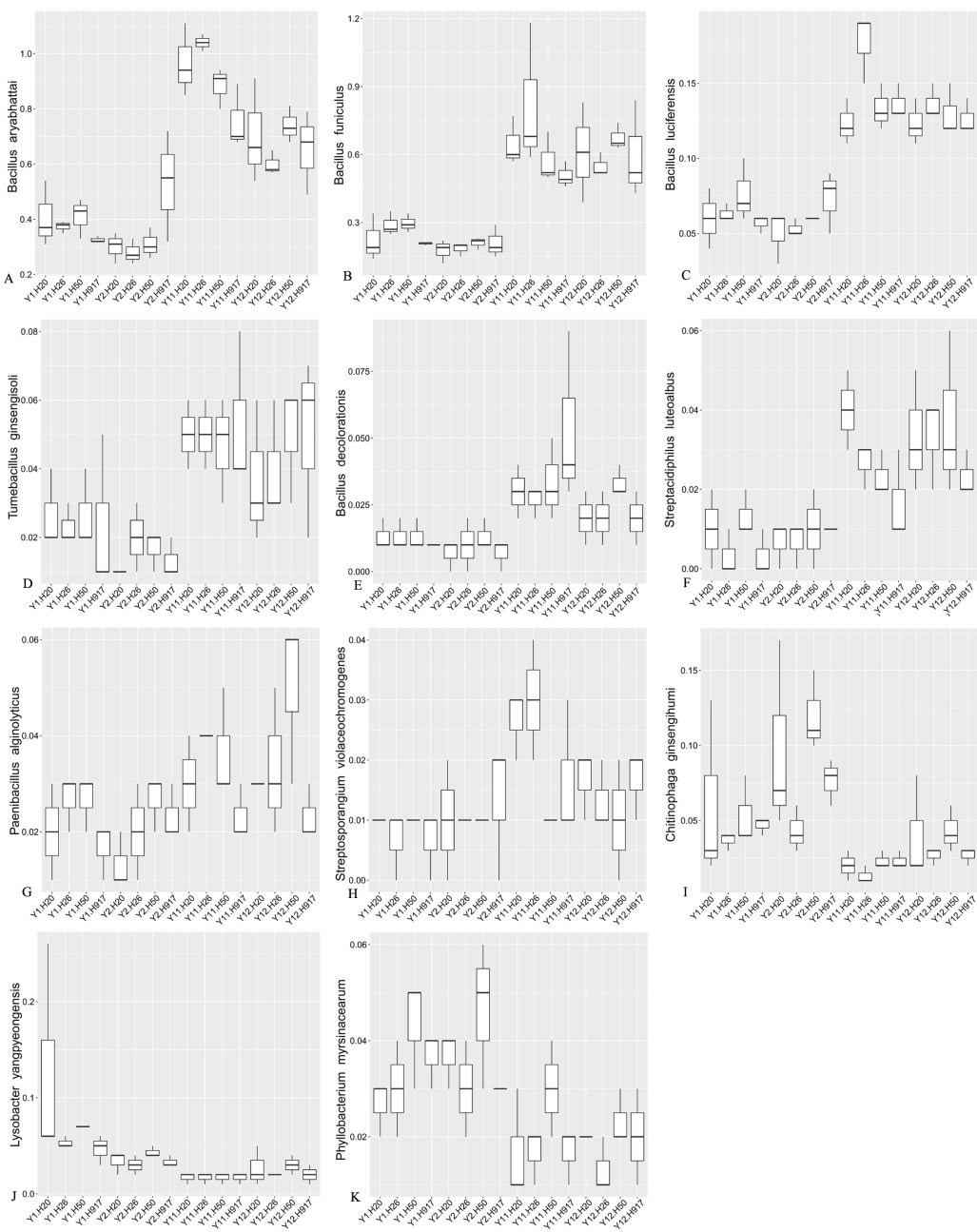

**Figure 6** Relative abundance of significantly changed taxa at species level in plots that were monocropped with peanuts for different periods (A–K).

monocropping plots were grouped into one class, and samples from 11- and 12-year plots were grouped into another.

Bacterial community functions also presented significant changes at different pathway levels across long-term monocropping. For KEGG pathways at KEGG level 2 that are involved in carbohydrate metabolism, endocrine system, excretory system, nucleotide metabolism, transport and catabolism, transcription, biosynthesis of other secondary

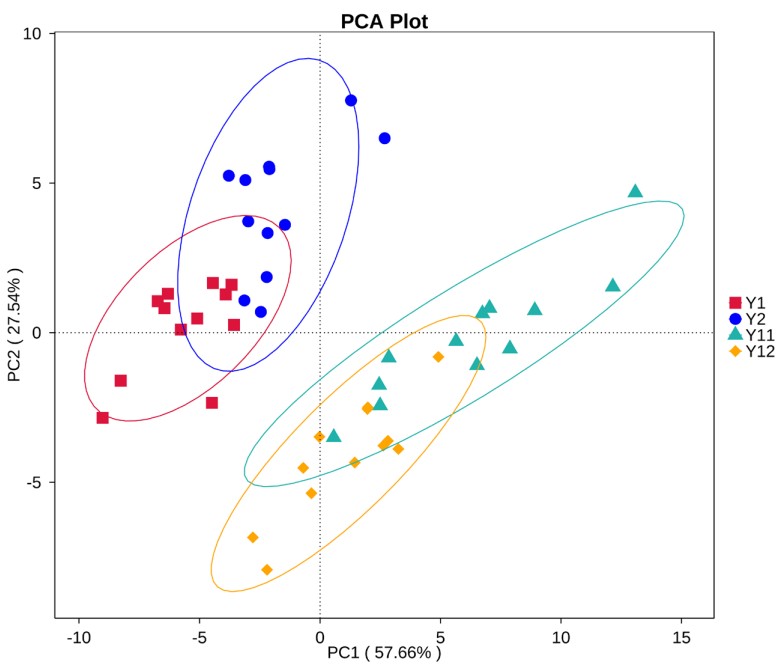

**Figure 7** Principal component analysis (PCA) of predicted bacterial functions at KEGG level 2 in short- and long-term monocropped peanut soils for different peanut varieties.

metabolites, and aging showed decreasing trends with long-term monocropping, but pathways affiliated with membrane transport and cellular community prokaryotes showed increased abundance (Table S4). There were 390 KEGG pathways at KEGG level 3 predicted in our libraries. The abundant pathways (>1.0%) that presented obvious changes in abundance with monocropping period were further examined. The pathways involved in DNA repair and recombination proteins, purine metabolism, transfer RNA biogenesis, exosome, amino acid related enzymes, pyrimidine metabolism, mitochondrial biogenesis, ribosome, oxidative phosphorylation, carbon fixation pathways in prokaryotes, aminoacyl tRNA biosynthesis as well as alanine, aspartate, and glutamate metabolism showed decreased abundance in long- compared to short-term monocropping samples. In contrast, pathways involved in ribosome biogenesis, messenger RNA biogenesis, RNA degradation, quorum sensing, glyoxylate and dicarboxylate metabolism, and lipid biosynthesis proteins were enriched in long-term monocropping samples (Table S4).

## DISCUSSION

It has been demonstrated that soil microorganisms play influential roles in the productivity and sustainability of agricultural systems (*Van der Heijden & Wagg, 2013*; *Vukicevich et al., 2016*). 'Soil sickness' caused by continuous peanut monocropping could be closely related to the dynamics of soil microbial communities. Our previous study using library analysis showed that soil microbial community structure shifted during three years of continuous peanut monocropping (*Chen et al., 2012*; *Chen et al., 2014*). Li et al. reported

that consecutive peanut monoculture changed communities of soil nematodes and fungi (*Li et al., 2014b*; *Li et al., 2015*). However, the specific dynamic succession of the bacterial community, which is most diverse in soil, under long-term peanut monocropping is not clear.

In the present study, we selected four peanut varieties with different monocropping responses and recorded the detailed feedback responses of their root soil bacteria communities to short- and long-term consecutive monocropping using high-throughput sequencing. Bacterial richness and diversity were measured by OTU number and Chao1 and Shannon indexes, as well as rarefaction curves. They all indicated that the majority of bacterial diversity was covered by the surveying effort and the diversity of the soil bacterial community generally declined with long-term peanut monocropping.

Our previous study showed that bacterial community structure presented significant dynamics during three years of peanut monocropping, but was less affected by peanut growth stage (*Chen et al., 2014*). *Li et al. (2014b)* reported that fungal communities were significantly selected by the monocropping period of peanut, but also were less affected by growth stage in the red soil region of southern China. In the present study, the UPGMA, PCA, and PCoA methods were used to cluster the bacterial communities within the 48 samples from short- and long-term monocropping soil. The analyses showed that all samples regardless of variety from 1- and 2-year monocropping plots were grouped into one class, and samples from 11- and 12-year plots were grouped into another. However, the cluster distance between samples from 1- and 2-year plots was shorter than that between samples from 11- and 12-year plots. Additionally, soil pH and contents of available P and K in soil also obviously changed with long-term monocropping of peanut. Monocropping time had a strong influence on the microbial communities as well as physico-chemical properties, but peanut variety and growth stage had little impact. Additionally, the soil bacterial community had significantly greater dynamics under long- compared to short-term monocropping time.

Our previous study demonstrated that bacterial communities at different taxonomic levels showed obvious dynamics during three years of peanut monocropping (*Chen et al., 2014*). Most of the obviously changed taxa at the order level showed abundance and diversity declines with monocropping time and only several taxa showed increased abundance and diversity (*Chen et al., 2014*). A third cropping of Jerusalem artichoke also decreased bacterial alpha-diversity compared to 1–2 years of monocropping (*Zhou et al., 2018*). In this study, under long-term peanut monocropping, the alpha-diversity of the soil bacterial community generally decreased. At the phyla level, nine taxa that accounted for 0.01–18.04% of the total sequences showed obvious decreases in abundance with monocropping time and generally their decreased abundance was accompanied by a decrease in diversity. Five taxa, representing 0.01–4.66% of the total sequences, showed obvious increases in abundance: the three relatively abundant taxa showed no obvious change in diversity with monocropping time, but that of the other two less abundant taxa showed increasing trends. It was suggested that simplification of bacterial communities is a common phenomenon during monocropping; however, some taxa increased their abundance and diversity, possibly due to their adaptability to a new microenvironment.

Soil microbe modifications could contribute to peanut soil sickness (*Chen et al., 2014*; *Li et al., 2014b*). *Li et al. (2012)* reported that bacteria proportion in total PLFA decreased from 67.4 to 53.0% in a peanut monocropping system, whereas the proportion of fungi increased from 16.9 to 32.8%. Consecutive peanut monocropping resulted in the selection of pathogenic and beneficial fungi (*Chen et al., 2012*; *Li et al., 2014b*). Soil nematode abundance and functional composition also changed with continuous peanut monocropping (*Li et al., 2015*). In our study, the bacteria community compositions and predicted functions at different levels all presented obvious changes with long-term peanut monocropping.

Some taxa that were identified at lower taxonomic levels, such as genus or species, showed significant changes in abundance with long-term peanut monocropping. At the genera level, *Steroidobacter*, *Candidatus Entotheonella*, *Nitrosospira*, *Pirellula*, *Piscinibacter*, *Ramlibacter*, *Lysobacter*, *Skermanella*, and *Planctomyces* showed decreased abundance with monocropping time. At the species level, *Chitinophaga ginsengihumi*, *Lysobacter yangpyeongensis*, and *Phyllobacterium myrsinacearum* also showed obvious decreases with long-term monocropping. Several taxa found were beneficial to plant growth. The genus *Nitrosospira* is reportedly related to nitrogen cycle progress (*Mellbye et al., 2017*) and *Candidatus Entotheonella* is related to biosynthesis (*Uria, Piel & Wakimoto, 2018*). Additionally, at the species level, *Phyllobacterium myrsinacearum* functions in nitrogen-fixing (*Gonzalez-Bashan et al., 2000*). Studies concerning roles in plants of the other decreased taxa at the genus or species levels were not found. However, this also demonstrated a decrease in abundance of the beneficial bacteria community with long-term peanut monocropping.

In contrast, some taxa at the genus or species levels showed increased abundance with long-term peanut monocropping. The genus *Bacillus* was dominant (2.22%) in our libraries, and increased their average abundance from 1.49% in 1- and 2-year to 2.96% in the 11- and 12-year libraries. However, the number of OTUs affiliated with the *Bacillus* genus in each sample (13–14 OTUs) was relatively low and changed little. This may suggest that identified members of *Bacillus* had good adaptability to the soil environment under long-term peanut monocropping. *Bacillus* species are reported as the most common biocontrol agents and have important traits such as plant growth-promoting properties (*Santoyo, Orozco-Mosqueda & Govindappa, 2012*; *Gomaa, 2012*). The dominant species *Bacillus aryabhattai* and *Bacillus funiculus* accounting for 0.58% and 0.42% of the total sequences, respectively, and the relatively abundant *Bacillus luciferensis* and *Bacillus decolorationis* accounting for 0.10 and 0.02% of the total sequences, respectively, all showed increased abundance with long-term monocropping. It was reported that *Bacillus aryabhattai* could improve growth of soybean, wheat, and *Xanthium italicum*, and could also improve mobilization and biofortification of zinc (*Lee, Ka & Song, 2012*; *Ramesh et al., 2014*). *Bacillus funiculus* is reportedly related to the degradation of sodium dodecyl sulfate (*Ajithkumar et al., 2003*). There are no reports concerning roles in plants of *Bacillus luciferensis* and *Bacillus decolorationis*. Additionally, several relatively abundant species, including *Tumebacillus ginsengisoli*, *Streptacidiphilus luteoalbus*, *Paenibacillus alginolyticus*,

and *Streptosporangium violaceochromogenes* also showed increased abundance with long-term peanut monocropping. However, no function-related studies of them were found. At the genus level, the functions of most of the increased taxa are unknown, but some studies claimed that *Nocardioides* species had roles in degradation of deoxynivalenol, 2,4-dinitroanisole, and melamine and its hydroxy derivatives (*Ikunaga et al., 2011*; *Takagi et al., 2012*; *Fida et al., 2014*). It was reported that allelochemicals from root exudates or decomposants of crops could induce autotoxicity and were closely related to soil sickness (*Asaduzzaman & Asao, 2012*; *Huang et al., 2013*). These allelochemicals accumulated with monocropping period, but not to high levels (*Yang et al., 2015*; *Li et al., 2014a*), possibly due to interactions between allelochemicals and soil microbes (*Li et al., 2014a*; *Wang et al., 2019*). Based on the analyses, the increased bacterial taxa that could promote plant growth or degrade potential soil allelochemicals should be closely related to soil remediation and may have potential application to relieve soil sickness under peanut monocropping.

Function prediction analyses showed significant changes in bacterial community functions at different pathway levels with long-term peanut monocropping. The analyses at KEGG levels 2 and 3 and KO level all showed similar clustering to that of bacterial community structure. Samples from short-term monocropping plots were grouped into one class, and samples from long-term plots were grouped into another. Comparing analyses of the abundance indicated that many detected KEGG pathways at KEGG levels 2 or 3 had obvious changes with long-term monocropping. Most had a decreasing trend with long-term peanut monocropping, and only a few showed an increase in abundance. Combined with the community structure variation analyses, both the bacterial community structure and function presented significant changes with long-term monocropping, and their simplification could be the main cause of soil sickness.

In conclusion, through tracking the detailed feedback responses of soil bacteria to long-term monocropping of four different peanut varieties, we provided field-based evidence that long-term monocropping could result in a general loss in bacterial diversity and remarkable changes in bacterial abundance and compositions as well as functions. Combined with our previous study (*Chen et al., 2014*), analyses in this study suggested that bacterial communities were obviously influenced by the monocropping period, but less influenced by peanut variety and growth stage. Additionally, soil pH and contents of available P and K in soil also obviously changed with long-term monocropping of peanut. Some bacterial taxa, with increased abundance have functions of promoting plant growth or degrading potential soil allelochemicals, should be closely related to soil remediation and may have potential application to relieve peanut soil sickness. A decrease in diversity and abundance of bacterial communities, especially beneficial communities, and simplification of bacterial community function with long-term peanut monocropping could be the main cause of peanut soil sickness. In the future, we will investigate dynamics of functional genes with long-term peanut monocropping using metagenomics. These studies will improve our understanding of the mechanism underlying peanut soil sickness.

### Funding

This study was supported by grants from The National Ten Thousand Youth Talents Plan of 2014 (W02070268), China Agriculture Research System (CARS-13), Taishan Scholar Project Funding, the National Natural Science Foundation of China (31701464), the Natural Science Fund of Shangdong Province (ZR2017YL017, ZR2016CM07), the Youth Scientific Research Foundation of Shandong Academy of Agricultural Sciences (2016YQN14), Agricultural Scientific and Technological Innovation Project of Shandong Academy of Agricultural Sciences (CXGC2016B02, CXGC2018E21), the Breeding Project from Department Science & Technology of Shandong Province (2017LZGC003) and the Basic Research Project of Qingdao (17-1-1-51-jch). The funders had no role in study design, data collection and analysis, decision to publish, or preparation of the manuscript.

### Grant Disclosures

The following grant information was disclosed by the authors:
The National Ten Thousand Youth Talents Plan of 2014: W02070268.
China Agriculture Research System (CARS-13).
Taishan Scholar Project Funding.
The National Natural Science Foundation of China: 31701464.
The Natural Science Fund of Shangdong Province: ZR2017YL017, ZR2016CM07.
Youth Scientific Research Foundation of Shandong Academy of Agricultural Sciences (2016YQN14).
Agricultural Scientific and Technological Innovation Project of Shandong Academy of Agricultural Sciences: CXGC2016B02, CXGC2018E21.
The Breeding Project from Department Science & Technology of Shandong Province (2017LZGC003).
The Basic Research Project of Qingdao (17-1-1-51-jch).

### Competing Interests

The authors declare there are no competing interests.

### Author Contributions

- Mingna Chen conceived and designed the experiments, performed the experiments, analyzed the data, prepared figures and/or tables, authored or reviewed drafts of the paper, and approved the final draft.
- Hu Liu performed the experiments, analyzed the data, authored or reviewed drafts of the paper, and approved the final draft.
- Shanlin Yu performed the experiments, authored or reviewed drafts of the paper, and approved the final draft.
- Mian Wang performed the experiments, prepared figures and/or tables, authored or reviewed drafts of the paper, and approved the final draft.
- Lijuan Pan analyzed the data, prepared figures and/or tables, authored or reviewed drafts of the paper, and approved the final draft.

- Na Chen, Tong Wang and Xiaoyuan Chi analyzed the data, authored or reviewed drafts of the paper, and approved the final draft.
- Binghai Du conceived and designed the experiments, authored or reviewed drafts of the paper, and approved the final draft.

## Data Availability

The data are available in the NCBI-Sequence Read Archive: PRJNA559575.

The raw data are availabile at Figshare:

Chen, Mingna; Du, Binghai (2019): RawData.rar. figshare. Dataset. https://doi.org/10.6084/m9.figshare.11351462.v1.

## Supplemental Information

Supplemental information for this article can be found online at http://dx.doi.org/10.7717/peerj.9024#supplemental-information.

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
