# Peer review of "Long-term continuously monocropped peanut significantly changed the abundance and composition of soil bacterial communities"

_PeerJ, doi:10.7717/peerj.9024_

## Round 0.1 · original submission · Major Revisions

Our reviewers have found several areas for improvement . Please address the reviewers' comments, especially those concerning the characterization of soil quality and plant growth, draining/irrigation/fertilization conditions, and whether your data refer to bulk soil or rhizosphere.

Reviewer 1 ·

Basic reporting

.

Experimental design

.

Validity of the findings

.

Additional comments

I appreciate the hard work on the manuscript entitled “Long-Term Continuously Monocropped Peanut Significantly Changed the Abundance and Composition of Soil Bacterial Communities”. The following comments would be helpful for this paper.
1. The object of this MS was focused on the changement of bacterial community based on the background of soil sickness caused by continuously monocropped. However, no related peanut sickness index was recorded in this MS. Meanwhile, I strongly suggested that more plant-growth data (e.g., weight, length, pod number) and the soil physichemical properities should be included. If not, the relationship between microbial community difference and the peanut soil sickness is so weak.
2. To the best of our knowledge, papers from authoritative journal have already suggested that the soil sickness in peanut continuously monocropped regime was mainly attribute to the imbalance of soil pathogenic fungal community. Hence, why the authors regard the difference of soil bacterial community assembly as the mainly reason of peanut soil sickness?
3. From the perspective of bio-informatic analyse, the presentation way of this MS was too traditional. I suggested that LefSE (LDA analysis) should be performed to further compare the significantly differential bacterial taxa among each treatment. Meanwhile, the PERMANOVA (Adonis) analyse should be calculated to further certify the significant level of community difference in PCA analysis.
4. In the terms of sampling processing, the authors have not clearly stated that the collecting soil samples is belonged to the peanut rhizosphere or the bulk soil.
5. As an nitrogen fixing crop, the community of rhizobia plays important role in the healthy growth of peanut. Since the MS only mentioned about the soil bacterial community, I suggested the authors should deeply discussed this problem.

Reviewer 2 ·

Basic reporting

The MS investigated the feedback responses of soil bacteria to short- and long-term continuous monocropping of four different peanut varieties using high-throughput sequencing techniques, and found Long-term consecutive monocropping could lead to a general loss in bacterial diversity and remarkable changes in bacterial abundance and composition, and inferred simplification of bacterial communities, especially beneficial communities, and bacterial community function simplification with long-term peanut monocropping could be the main cause of peanut soil sickness. It is a good MS, and some suggestions are expected to help further improve the quality of manuscripts.
It is better to simplify the results descriptions (including figures ) and enhance bacterial function discussions (beneficial or pathogenic ). Some merging of unimportant types or without specific functions will increase the readability since it was not clear in the current version.
Line 94, I am not sure ”this is the first study”. It perhaps occurred in a specific region or soil condition.
Since varieties had little effect on soil microorganisms, what is the mechanism of sensitive or tolerant peanuts to monocropping?

Experimental design

Line 105, what is mean 1m high ? What’s kind of soil ? How to drain on rainy days?
Was one peanut variety continuously planted for 11 or 12 years without variety change?
Besides soil microorganisms, main physical and chemical properties of soil should be list in a Table(S)

Validity of the findings

no comment

Reviewer 3 ·

Basic reporting

Chen’s manuscript entitled “Long-Term Continuously Monocropped Peanut Significantly Changed the Abundance and Composition of Soil Bacterial Communities” investigated the response of soil bacterial community to the peanut variety, growth stage and continuous cropping time, and found that the microbial communities were changed among continuous cropping times. The authors attributed the soil sickness caused by continuous monocropping to the declined abundance of the beneficial bacterial taxa.

Experimental design

From which rhizocompartment the soil was collected? Need a more detailed description of how and why the soil sample was taken.

Validity of the findings

It is still hard to find a clear conclusion of the manuscript. The structure of the soil microbiome has changed over time, this could be due to too many factors, such as raining, irrigation, fertilization et al. What is the main reason for the changes in microbial community structure should be discussed.

Additional comments

The language of the manuscript need to be revised by a native English speaker. There is a lot of confusing statements and mistakes of the basic concepts in the manuscript. For instance, the first sentence of the Abstract “Soil sickness is the progressive loss of soil quality due to continuous monocropping”. Soil sickness could be introduced by a number of practices other than monocropping.

---

## Round 0.2 · Minor Revisions

Please address the remaining issues. Please be more thorough than originally in your replies to reviewer #3

Reviewer 2 ·

Basic reporting

no comment

Experimental design

no comment

Validity of the findings

no comment

Additional comments

Line 187: “The soil pH in samples from 2-year monocropped plots (7.20 - 8.05) were slightly higher than that in 1-year monocropped plots samples (8.30 - 8.59)”, Check it, please.
What is “Y1.H20…..Y12H917”? It should be described when the codes first emerged.
Line 357 “These analyses indicated that soil pH, available P, K could be closely related with soil bacterial communities under long-term monocropping of peanut”, but there was no description in the MS.

Reviewer 3 ·

Basic reporting

I still strongly suggest the language be revised by a native English speaker. For instance,in line 138, "distance" is very confusing, do the authors mean "depth" ?

Experimental design

It is still not possible to figure out from which rhizocompartment the soil was collected? rhizosphere? rhizoplane or bulk soil?

Validity of the findings

It is still hard to figure out a clear conclusion of the manuscript mainly due to the language issue. For example, the term "simplification of bacterial community", which has been described as the main reason for the peanut soil sickness, is very confusing.

Additional comments

Most of my comments in the first round review have not NOT been addressed.

---

## Round 0.3 · accepted · Accept

I am satisfied with your responses to the reviewers' comments.

Reviewer 2 ·

Basic reporting

no comment

Experimental design

no comment

Validity of the findings

no comment